# Associations Between Electronic Vapor Product Use and Prescription Opioid Misuse Among High School Students in the United States; A Retrospective Cross-Sectional Analysis

**DOI:** 10.3390/children12111476

**Published:** 2025-11-01

**Authors:** Killian M. Pache, Lionel Kameni, Cornelius B. Groenewald

**Affiliations:** 1School of Medicine, Case Western Reserve University, Cleveland, OH 44106, USA; kmp178@case.edu; 2Department of Anesthesiology, Perioperative and Pain Medicine, Stanford University School of Medicine, Stanford, CA 94309, USA; leokmen@stanford.edu

**Keywords:** electronic vapor product use, adolescent, opioid misuse, high school students, e-cigarette use

## Abstract

**Highlights:**

**What are the main findings?**
•The use of electronic vapor products (EVPs) in adolescents was found to be correlated with increased current opioid misuse•This relationship had a dose-dependent effect, with an increased frequency of EVP use correlating with an increased prevalence of current opioid misuse

**What is the implication of the main finding?**
•Adolescents with EVP use may be at increased risk for opioid misuse and should be identified for targeted screening and education•Public health measures aiming to reduce EVP use in adolescents may want to highlight an increased risk of opioid misuse as a potential sequela of EVP use initiation

**Abstract:**

**Study objective:** Electronic vapor product (EVP) use remains prevalent among adolescents in the United States. EVP use may be associated with high-risk substance use behaviors. This study investigates the association between current EVP use and current opioid misuse in high school students. **Methods:** We conducted a secondary analysis of the 2021 Youth Risk Behavior Survey (YRBS) (*n* = 7471). We first examined the prevalence rates of current prescription opioid misuse between participants ages 14 to 19 who reported any EVP use compared to those who did not report any EVP use. We conducted an adjusted Poisson regression analysis to determine whether EVP use was associated with prescription opioid misuse after controlling for multiple variables, including age, biological sex, race, ethnicity, alcohol use, other tobacco use, marijuana use, and depressive symptoms. **Results:** Among participants, 17.2% (95% confidence interval (CI): 15.4–19.1%) reported current EVP use. The prevalence of current opioid misuse was higher in the group of students who use EVPs (13.3%) as compared to students who do not use EVPs (3.2%) (*p* < 0.0001). Compared to participants who did not use EVPs, those with EVP use had an 80% greater prevalence of opioid misuse, after adjusting in regression analyses (adjusted prevalence rate ratio: 1.8, 95% CI: 1.4–2.3, *p* < 0.0001). **Conclusions:** Our findings highlight the significantly increased prevalence of opioid misuse among this population and the need to educate adolescents about the additional risks associated with nicotine and EVP use. Longitudinal studies are needed to test causal relationships and better understand the biobehavioral mechanisms that connect EVP use and opioid misuse.

## 1. Introduction

Tobacco product use remains a significant public health concern. In the United States, the death toll attributable to tobacco-smoking-related illnesses such as lung cancer, chronic obstructive pulmonary disease (COPD), and cardiovascular disease is estimated at 490,000 annually [1]. Tobacco control measures have secondarily resulted in a significant decrease in adolescent cigarette smoking, from 27.5% in 1991 to 3.8% in 2021 [2]. However, electronic vapor products (EVPs), including e-cigarettes, vapes, vape pens, e-cigars, e-hookahs, and hookah pens were developed and subsequently entered the United States marketplace around 2007 [3]. EVPs deliver tobacco products in aerosol form, using battery-operated devices. Originally marketed as a safer alternative to tobacco smoking and as a cessation aid for adults who smoke, the use of EVPs has significantly increased among adolescents over the past two decades [4]. The most recent estimates from the Youth Risk Behavior Survey (YRBS) found that 36% of high school students have tried EVPs, and that 18% of high school students have used an EVP in the last 30 days. The high prevalence of EVP use among adolescents (defined as people between the ages of 10 and 19) has raised concerns that it may act as a primer for other risky substance use behaviors [4].

Previous research in adults found strong associations between tobacco smoking and opioid misuse [5], and adults with substance use disorders are much more likely to smoke cigarettes than the general public [6]. Indeed, nicotine and opioid addictions share overlapping neural pathways that can be mutually reinforcing: activating nicotinic acetylcholine receptors may increase the rewarding action of mu-opioid receptors, and vice versa [7]. However, despite these known associations, and the rapidly increasing popularity of electronic vapor products among adolescents, the associations between EVP and opioid misuse behaviors among adolescents remain poorly described.

This study used nationally representative data from the 2021 Youth Risk Behaviors Survey (YRBS) to determine associations between current electronic vapor product use and current opioid misuse among high school students in the United States. We hypothesized that adolescents with current EVP use would be more likely to also report current prescription opioid misuse, as compared to adolescents who did not use EVPs. Moreover, we hypothesize that these associations remain significant after controlling for other known risk factors of both EVP and opioid misuse, including alcohol use, marijuana use, depressive symptoms, and sleep insufficiency. Our secondary aim was to determine whether a dose-dependent relationship exists between EVP use (number of days used in the past month) and the prevalence of opioid misuse in adolescents. A better understanding of the relationship between EVPs and opioid misuse may improve screening, prevention, and supportive care for adolescents who use EVPs.

## 2. Materials and Methods

We conducted a cross-sectional analysis of the 2021 Youth Risk Behavior Survey (YRBS). The YRBS is a self-administered biennial survey distributed by the Center for Disease Control and Prevention to high school students (grades 9 through 12) across the United States. Surveys are self-administered in the classroom after parents give permission. Time to complete the survey is 45 min. The YRBS is nationally representative and is the largest public health surveillance system of adolescent health behavior risk factors in the United States. The overall response rate for the 2021 YRBS was 57.5%, which is consistent with previous years. Additional information about the YRBS is available in existing publications [8] and online at https://www.cdc.gov/healthyyouth/data/yrbs/index.htm (accessed on 15 April 2025). As data is de-identified and publicly available, our IRB deemed this study as exempt from review.

### 2.1. Study Population

The total sample size for participants who answered survey questions about opioid misuse was 9866 in the 2021 YRBS. Missing data was analyzed using listwise deletion as imputation of missing data is not recommended for YRBS. Therefore, the final data set was limited to participants with all available data, which resulted in a final sample of 7471 participants.

### 2.2. Measures

#### 2.2.1. Electronic Vapor Product Use

Current EVP use was evaluated with the question: “During the past 30 days, on how many days did you use an electronic vapor product?” with the option choices of “0 days”, “1 or 2 days”, “3 to 5 days”, “6 to 9 days”, “10 to 19 days”, “20 to 29 days”, and “All 30 days”. Participants who reported EVP use during the past 30 days were classified as adolescents who currently use EVPs. Similarly, current combustible cigarette use was evaluated with the question: “During the past 30 days, on how many days did you smoke cigarettes?” with the option choices of “0 days”, “1 or 2 days”, “3 to 5 days”, “6 to 9 days”, “10 to 19 days”, “20 to 29 days”, and “All 30 days”. Participants who reported smoking combustible cigarettes but denied EVP use during the past 30 days were classified as current “other tobacco” users.

#### 2.2.2. Prescription Opioid Misuse

Current prescription opioid misuse was evaluated with the question: “During the past 30 days, how many times did you take prescription pain medicine without a doctor’s prescription or differently than how a doctor told you to use it?” With answer choices of “0 times”, “1 or 2 times”, “3 to 9 times”, “10 to 19 times”, “20 to 39 times”, and “40 or more times”. Participants were specifically asked to include medications such as codeine, Vicodin, OxyContin, Hydrocodone, and Percocet. The YRBS then categorized those participants reporting misusing opioids one or more times during the past 30 days as people who are currently misusing prescription opioids.

#### 2.2.3. Covariates

Covariates that we identified before data analysis as having been associated with both EVP use and prescription opioid misuse included alcohol use [9], marijuana use [9], and depressive symptoms [10]. We included these covariates as well as demographic information in our multivariate regression analysis.

**Sociodemographic.** Students were asked to self-report their age, sex, and race and ethnicity (White, non-Hispanic; Black, non-Hispanic; Hispanic; Asian, non-Hispanic; American Indian/Alaska Native; Native Hawaiian/Other Pacific Islander; and multiple race)

**Alcohol and marijuana use.** Participants who reported one or more episodes of alcohol use over the past 30 days were coded as having current alcohol use; while those reporting marijuana use more than once over the past 30 days were coded as having current marijuana use. Both variables were coded as binary (0 = No; 1 = Yes).

**Depressive symptoms.** Depressive symptoms were captured as a binary (0 = No; 1 = Yes) variable in response to the question “During the past 12 months, did you ever feel so sad or hopeless almost every day for two weeks or more in a row that you stopped doing some usual activities?”. Of note, binary coding of the alcohol use, marijuana use, and depressive symptoms was conducted by YRBS and corresponds to the variables QN41, QN47, and QN25, respectively.

#### 2.2.4. Study Sample and Data Analysis

Analyses were conducted using Stata version 18.0 (StataCorp College Station, TX, USA); α was set at 0.05, and hypothesis testing was two-tailed. Missing data were not imputed, and case-wise deletion was used in our statistical models. We adjusted for the complex sample design of YRBS by using sampling weights, regional stratification, and primary sampling unit information to provide nationally representative estimates of high school students in the United States. We conducted descriptive analysis to analyze the distribution of demographic variables, comparing those with EVP use to those without EVP use.

To address our primary aim, we first conducted Rao-adjusted Pearson chi-square analysis to compare prevalence rates of current prescription opioid misuse (binary outcome) between participants who reported any EVP use to those who did not report any EVP use. We then conducted adjusted Poisson regression analysis, reported as prevalence rate ratios, to determine whether EVP use was associated with prescription opioid misuse after controlling for multiple variables, including age, biological sex, race, ethnicity, alcohol use, other tobacco use, marijuana use, and feeling sad/depressed. We also conducted multivariate poisson regression using current prescription opioid misuse as an ordinal outcome (with options “0 times”, “1 or 2 times”, “3 to 9 times”, “10 to 19 times”, “20 to 39 times”, and “40 or more times”.) in our models to determine whether EVP use was associated with it.

To address our secondary aim, we first conducted Rao-adjusted Pearson chi-square analysis to determine whether prevalence rates of current prescription opioid misuse increased as the level of past-month EVP use increased among participants. We then conducted adjusted Poisson regression analysis, reported as prevalence rate ratios, to determine whether higher levels of EVP use were associated with prescription opioid misuse after controlling for multiple variables, including age, biological sex, race, ethnicity, alcohol use, other tobacco use, marijuana use, and feeling sad/depressed

## 3. Results

Our sample included 7471 participants, of whom 17.2% (95% CI: 15.4–19.1%) reported current EVP use. EVP use was more prevalent among older students as compared to younger students, with 40.7% of students who currently use EVPs being 17 or older despite representing 30.2% of the students who responded to the survey. Females were more likely to report EVP use relative to males, as 55.8% of students who used EVPs were female despite representing 47.4% of the total students who completed the survey. EVP use was more common among those students who identified as being White, non-Hispanic (60.8% of students who currently use EVPs and 52.9% of the total respondents) and American Indian/Alaskan Native and Native Hawaiian/other Pacific Islander (1.2% of students who currently use EVPs and 0.7% of the total respondents) as compared to other racial and ethnic groups. EVP use was much more common among students who reported marijuana use (57.0% of students who currently use EVPs and 13.7% of the total respondents) and among those with current alcohol use (68.6% of students who currently use EVPs and 21.2% of the total respondents). Students who reported feelings of sadness or loneliness also were more likely to use EVPs (66.0% of students who currently use EVPs and 41.0% of the total respondents). Full characteristics of the study sample by EVP use are presented in Table 1.

The prevalence of current opioid misuse was significantly higher among students who reported EVP use (13.3%) as compared to students who denied EVP use (3.2%) (*p* < 0.0001). This difference remained statistically significant in our multivariable models: EVP use increased the adjusted prevalence of prescription opioid misuse by 80% (adjusted prevalence rate ratio (aPR): 1.8, 95% CI: 1.4–2.3, *p* < 0.001) (Table 2). In our models using opioid misuse as an ordered outcome, we found that adolescents with EVP use had 80% increased risk of reporting higher levels of opioid misuse compared with those in lower categories of opioid misuse combined (aPR: 1.8, 95% CI: 1.4–2.5, *p* < 0.0001).

In our sample, 3.8% of participants used EVPs on one or two days during the past month, while 7.4% used EVPs between 3 and 29 days, and 5% used EVPs daily during the past 30 days. Examining the dose-dependent relationship between EVP use and current prescription opioid misuse revealed that the prevalence of prescription opioid misuse increased as students reported increased EVP use during the past month (Figure 1). The prevalence of prescription opioid misuse among participants with 1–2 days of EVP use was 9.7% but increased to 16.8% among those with daily EVP use. In our adjusted model, the risk for opioid misuse increased as the level of past-month EVP use increased. Participants who reported 1–2 days of EVP use had an 80% increased risk of opioid misuse (aPR: 1.8; 95%CI: 1.3–2.3), while those who reported EVP use every day had an 130% greater risk of reporting opioid misuse (as compared to those reporting no EVP use) (aPR: 2.3; 95% CI: 1.5–3.5) (Table 3).

## 4. Discussion

We used data from the nationally representative 2021 Youth Risk Behavior Survey to examine associations between EVP use and prescription opioid misuse among high-school students in the United States. As our primary aim, we found that students who reported EVP use during the past 30 days were more likely to report current prescription opioid misuse compared to those who were not currently using EVPs. This association between EVP use and opioid misuse remained highly significant after controlling for multiple covariates, including sociodemographic factors and other substance use behaviors. Additionally, as our second aim, we examined the dose-dependent relationship between frequency of current EVP use and prescription opioid misuse. We found that an increased use of EVPs was strongly associated with an increased prevalence of prescription opioid misuse.

While previous research has investigated the association between EVP use and substance misuse among adults, the goal of this study was to understand this association in the adolescent population. Studying adolescents is critical, because this is the time of life when substance use behaviors start [11]. Moreover, our work is particularly relevant in the context of the current opioid epidemic, as opioid misuse in adolescents has been linked to increased risk for morbidity and mortality via opioid overdose [12]. This poses an important challenge for this vulnerable group and underscores the need for targeted public health interventions in adolescent EVP and opioid misuse. A strong body of research suggests a multi-pronged approach which entails early identification of at-risk adolescents as well as education on the potential harm of EVP use [13]. By identifying at-risk adolescents early, resources can be focused and appropriately distributed to maximize harm reduction.

Studies of adult populations have repeatedly found that individuals who engage in nicotine and cigarette use are at an elevated risk for misusing opioids [14]. A large meta-analysis published in 2019 with over 175,000 participants found that the pooled odds ratio of opioid misuse disorder was 8.23 (95% CI: 3.07–22.09) for adults who currently smoke compared to those who did not. Additionally, the meta-analysis investigated the relationship between age of smoking onset and opioid misuse and found that an early onset of smoking, defined as being under 14 years old at smoking onset, was positively associated with opioid misuse and opioid misuse disorder [14]. This finding adds evidence to the theory that exposure to nicotine at a younger age may increase vulnerability to the misuse of other substances, such as opioids. Other studies that have focused on individuals with opioid misuse disorders have shown an exceedingly high rate of nicotine use; for example, one study of patients with opioid misuse disorder enrolled in methadone treatment found that over 85% of the patients were smoking cigarettes [15]. One proposed mechanism that may contribute to the association between nicotine use and opioid misuse is an increased prevalence of chronic pain among those who use nicotine. Multiple studies investigating chronic pain have found that nicotine use and dependence were strong predictors of prescription opioid misuse [16,17]. One study investigating predictors for low back pain in adolescents found that one of the strongest factors associated with hospitalization due to low back pain was daily smoking [18]. However, previous studies investigating this relationship between EVP use and opioid misuse have mainly focused on adult populations, and the studies investigating adolescent populations have not been nationally representative.

Current biopsychosocial evidence points to an interplay of various factors during adolescents’ critical developmental stage that may influence social behaviors as well as nicotine’s effects on the brain. The endogenous opioid system is hypothesized as an important neurobiological pathway in the addictive properties of nicotine. Nicotine produces its effects through the activation of nicotinic acetylcholine receptors, with downstream upregulation of mu opioid receptors, which play a critical role in addiction [7,19]. As such, chronic and long-term use of nicotine leading to upregulation of mu-opioid receptor expression could provide a physiological basis for increased sensitivity to opioid misuse and addiction among people who use EVPs. The inverse of this relationship can be seen in the use of the opioid receptor antagonists, naltrexone and bupropion, decreasing nicotine withdrawal symptoms and nicotine cravings in some patients [19,20]. In addition to its effects on the endogenous system, nicotine from EVPs can also affect the brain’s reward pathways, thereby making adolescents more prone to using other substances like opioids that provide similar pleasurable effects [7,19]. Nicotine or EVP use in adolescent populations may be especially impactful, as biological and signaling pathway changes are occurring in a developing brain that does not have a fully formed frontal cortex with strong executive function to regulate decision-making and risk-assessment. These changes may affect adolescents’ vulnerable brain development, leading to behavioral impairment. Behavioral impairment can manifest as poor impulse control with a priming effect for subsequent increased adolescent participation in health-risk behaviors [21]. Social dynamics must also be considered in the context of adolescent decision making, as adolescents are particularly susceptible to peer pressure, which can be an external factor on the likelihood of engaging in risky behaviors, including substance use. Taken together, the confluence of these biological pathways, brain development changes, and social factors affecting adolescents all provide possible explanations underlying links between adolescents’ EVP use and prescription opioid misuse.

The findings of this study must be interpreted in context of several limitations. First, the cross-sectional survey captures data from a single point in time which does not allow us to draw conclusions about the causal relationship between EVP use and current opioid prescription misuse. However, our dose-dependent finding between EVP use and opioid misuse aligns with the results of studies conducted among adults which found a stronger relationship between opioid misuse among dependent adults who smoked heavily compared to those who smoked less heavily [16]. Finding a dose-dependent relationship between these variables is significant as it adds evidence to support a causal relationship between two variables by fulfilling one of Bradford Hill’s criteria for establishing causality in epidemiology. Importantly, this does not prove causation, however, as there may be additional confounding variables, and different research methodology would be needed to confirm this. One way to build upon this finding and add evidence to the possibility of this being a causal relationship would be to establish the temporal order between EVP use and opioid misuse, which this data set does not do. Longitudinal data captured over the course of students’ high school experiences would provide valuable insight into the directionality and temporality of this relationship. Another limitation to this study is that opioid misuse is measured by only one question that does not capture differences in potential routes of opioid misuse or extent of misuse. A more comprehensive measure of opioid misuse that captured frequency, quantity, and method of consumption, for example, would provide more data and allow for a more nuanced understanding of this relationship. The choice of which screening tool to use in evaluating opioid misuse in adolescents has been shown to have a significant effect on the reported rates of opioid misuse and should be considered carefully in any investigation into this topic [22]. The integration of validated pediatric screening tools, such as the CRAFFT Screening Tool, could improve opioid misuse data collection. Stigma surrounding opioid misuse may result in the reported YRBS rate being lower than the actual rate of opioid misuse. Similar limitations in the depth of questions and self-report status also apply to other variables measured such as marijuana use, alcohol use, and depressive symptoms. Despite these limitations, this study consists of a nationally representative sample of high school students and contributes valuable information to the field of adolescent addiction medicine. The large and diverse sample of adolescents allows for a generalizable understanding of the relationship between EVP use and prescription opioid misuse at a national level in the United States.

Recognizing the existence of a potential relationship between EVP use and adolescent opioid misuse could have significant public health implications by improving adolescent education, identifying adolescents at higher risk for opioid misuse, and improving resource allocation to more effectively prevent harm. While the popularity of cigarettes in adolescents has continued to fall, many adolescents still consider EVPs to be a safer alternative [23]. This problem is compounded by EVP marketing, which appeals to adolescents with sweet and tropical flavors, as well as the increased potential for peer pressure due to EVPs’ current popularity. Public health strategies to combat EVP misconceptions should be initiated through efforts to spread awareness of the health risks and potential sequelae of EVP use initiation. Similar public health measures targeting cigarette use have had immense success in reducing adolescent cigarette conception. Local efforts should include school and community-based programs with the active participation of teachers, parents, school counselors, healthcare providers, and other adolescents. On a larger scale, policy makers should advocate for public education through mass media or public health campaigns that highlight the potential harm of EVPs. Finally, consistently screening adolescents for EVP use during well-child visits could open the door for focused follow-up opioid screening and create an opportunity to provide education on the risks of EVPs.

## 5. Conclusions

In summary, current EVP use among adolescents in the United States was associated with an increased risk for current prescription opioid misuse. Additionally, increased EVP usage within the past month was associated with a higher risk for current opioid misuse. This relationship could be seen with the increasing prevalence of opioid misuse among adolescents who used EVPs daily compared to those who used EVPs less frequently. This finding suggests the possibility of a dose-dependent relationship existing between EVP use and opioid misuse. The results of this study should be considered when educating adolescents on the risks of nicotine and EVP use, and pediatricians and parents should consider these results when screening for potential warning signs of opioid misuse. The findings of this study highlight the need for further research in understanding the relationship between adolescent EVP use and opioid misuse. Given that causal relationships cannot be established from our cross-sectional analysis, a future longitudinal study with improved opioid misuse screening questions among adolescents would provide valuable added insight and clarity into this important relationship.

## Figures and Tables

**Figure 1 children-12-01476-f001:**
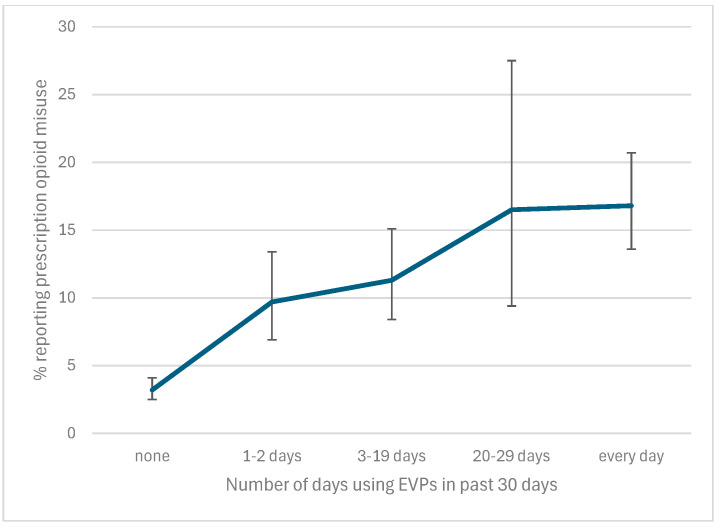
Prevalence of prescription opioid misuse by the number of days students reported using EVPs within the past 30 days.

**Table 1 children-12-01476-t001:** Demographic variables describing the age, gender, race and ethnicity, substance use pattern, and depressive symptoms for study participants as a whole group and divided between EVP use and non-use.

	Total (n = 7471)		EVP Use (n = 1367)		No EVP Use (n = 6104)		*p*-Value
Variable	Weighted %	95% CI	Weighted %	95% CI	Weighted %	95% CI	
**Age**							
14 years	19.2	[17.7, 20.9]	12.7	[9.8, 16.5]	20.6	[18.8, 22.5]	0.000
15 years	24.8	[23.2, 26.4]	20.6	[17.9, 23.6]	25.6	[24.0, 27.3]	
16 years	25.8	[24.4, 27.3]	25.9	[22.6, 29.5]	25.8	[24.2, 27.5]	
17 years	24.0	[22.5, 25.6]	31.3	[27.8, 35.1]	22.5	[21.1, 23.9]	
18 years or older	6.2	[5.3, 7.2]	9.4	[7.0, 12.5]	5.5	[4.7, 6.4]	
**Female**							
Male	52.6	[50.9, 54.3]	44.2	[41.2, 47.2]	54.3	[52.6, 56.1]	0.000
Female	47.4	[45.7, 49.1]	55.8	[52.8, 58.8]	45.7	[43.9, 47.4]	
**Race and ethnicity**							
White, non-Hispanic	52.9	[48.1, 57.5]	60.8	[56.1, 65.3]	51.2	[46.2, 56.2]	0.000
Black, non-Hispanic	9.9	[7.4, 13.1]	7.2	[5.3, 9.6]	10.4	[7.7, 14.0]	
Hispanic	9.4	[7.5, 11.8]	8.4	[5.7, 12.2]	9.6	[7.8, 11.9]	
Asian	5.6	[3.3, 9.4]	1.4	[0.9, 2.3]	6.5	[3.8, 10.8]	
Am Indian/Alaska Native	0.7	[0.5, 1.0]	1.2	[0.8, 1.8]	0.6	[0.4, 0.9]	
Native Hawaiian/Other PI	0.3	[0.2, 0.4]	0.4	[0.1, 1.5]	0.3	[0.2, 0.4]	
Multiple	21.2	[19.3, 23.3]	20.5	[17.7, 23.6]	21.4	[19.3, 23.6]	
**Current tobacco (other than EVP) use**							
No	99.3	[99.0, 99.4]	100	[99.2–100]	99.1	[98.9, 99.3]	
Yes	0.8	[0.6, 1.0]	0	[0.0–0.8]	0.9	[0.7, 1.2]	
**Current marijuana use**							
No	86.3	[84.5, 87.9]	43.0	[39.0, 47.1]	95.3	[94.1, 96.2]	0.000
Yes	13.7	[12.1, 15.5]	57.0	[52.9, 61.0]	4.7	[3.8, 5.9]	
**Current alcohol use**							
No	78.8	[77.0, 80.5]	31.4	[27.5, 35.7]	88.7	[87.7, 89.6]	0.000
Yes	21.2	[19.5, 23.0]	68.6	[64.3, 72.5]	11.3	[10.4, 12.3]	
**Felt sad or hopeless**							
No	59.0	[57.1, 60.8]	34.0	[30.3, 37.8]	64.1	[62.2, 66.0]	0.000
Yes	41.0	[39.2, 42.9]	66.0	[62.2, 69.7]	35.9	[34.0, 37.8]	

**Table 2 children-12-01476-t002:** Prevalence and adjusted prevalence ratios of opioid misuse among adolescents who currently use EVPs and those who do not, along with potential confounding demographic variables from Table 1.

	Prevalence of Opioid Misuse			Adjusted Prevalence Ratios		
Variable	Weighted %	95% CI	*p*-Value	aPR *	95% CI	*p*-Value
**Current electronic vapor use**						
No	3.2	[2.5, 4.1]	<0.0001			
Yes	13.3	[11.5, 15.4]		1.8	[1.4, 2.3]	<0.0001
**Age**						
14 years	5.8	[4.5, 7.5]	0.609			
15 years	4.8	[3.6, 6.4]		0.7	[0.5, 1.1]	0.123
16 years	4.6	[3.2, 6.7]		0.6	[0.4, 0.9]	0.014
17 years	4.6	[3.4, 6.2]		0.6	[0.4, 0.8]	0.003
18 years or older	5.4	[3.9, 7.6]		0.7	[0.4, 1.1]	0.124
**Female**						
Male	3.1	[2.4, 3.9]	<0.0001			
Female	7.0	[5.7, 8.6]		1.5	[1.1, 2]	0.004
**Race and ethnicity**						
White, non-Hispanic	3.9	[3.2, 4.8]	0.004			
Black, non-Hispanic	7.2	[4.6, 11.1]		2.1	[1.3, 3.6]	0.006
Hispanic	5.9	[4.8, 7.4]		1.7	[1.2, 2.4]	0.008
Asian	3.6	[2.2, 5.9]		1.4	[0.9, 2.4]	0.17
Am Indian/Alaska Native	7.0	[2.3, 19.4]		1.3	[0.5, 3.8]	0.571
Native Hawaiian/Other PI	12.5	[5.1, 27.5]		2.9	[1.2, 6.6]	0.015
Multiple	6.2	[4.6, 8.4]		1.5	[1.1, 2]	0.01
**Current tobacco (other than EVP) use**						
No	4.9	[4.1, 5.8]	0.193			
Yes	12.6	[2.7, 42.5]		2.3	[0.5, 10]	0.273
**Current marijuana use**						
No	3.4	[2.8, 4.1]	0.000			
Yes	14.7	[12.4, 17.4]		1.7	[1.2, 2.3]	0.002
**Current alcohol use**						
No	3.1	[2.5, 3.9]	<0.0001			
Yes	11.7	[10.2, 13.4]		1.8	[1.4, 2.4]	<0.0001
**Felt sad or hopeless**						
No	2.1	[1.4, 3.0]	<0.0001			
Yes	9.1	[7.8, 10.4]		2.6	[1.8, 3.8]	<0.0001

* Adjusted for age, sex, race/ethnicity, current tobacco (other than EVP) use, current marjuana use, current alcohol use, and depressive symptoms. aPR = adjusted prevalence ratios, CI = confidence interval.

**Table 3 children-12-01476-t003:** Multivariable associations between intensity of past month electronic vapor product (EVP) use and prevalence rate ratio of opioid misuse.

Number of Days in Past Month of EVP Use	aPR *	95% CI	*p*-Value
0 days	ref		
1 or 2 days	1.8	[1.3, 2.3]	<0.0001
3–19 days	1.5	[1, 2.2]	0.078
20–29 days	2.2	[1.4, 3.5]	0.001
every day	2.3	[1.5, 3.5]	0.001

* Adjusted for age, sex, race/ethnicity, current tobacco (other than EVP) use, current marijuana use, current alcohol use, and depressive symptoms. aPR = adjusted prevalence ratios, CI = confidence interval.

## Data Availability

YRBS Data Set for download: https://www.cdc.gov/yrbs/data/index.html (accessed on 28 November 2023).

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
