# Peer review of "Associations Between Electronic Vapor Product Use and Prescription Opioid Misuse Among High School Students in the United States; A Retrospective Cross-Sectional Analysis"

_children, 2025, doi:10.3390/children12111476_

Round 1
Reviewer 1 Report
Comments and Suggestions for Authors
Associations between electronic vapor products use and prescription opioid misuse among high school students in the United States; a retrospective cross-sectional analysis
COMMENTS FROM THE REVIEWER
General Comments
This manuscript assesses a crucial topic in public health, particularly concerning the opioid epidemic in the United States and the rising use of vaping (nicotine) products among a vulnerable group: adolescents. It highlights an underexplored association among the school population and presents evidence from a nationally representative survey. The objectives are clearly defined, the methodology is explained, and the results are complemented by tables and figures that enhance understanding.
Specific comments
The analysis of the study enhances our understanding of substance use among adolescents. However, some methodological limitations need clearer emphasis to enhance the manuscript's overall quality for publication in a peer-reviewed journal:
- Cross-sectional analysis and Causal relationship: The use of cross-sectional analysis limits the establishment of causal relationships between PVE and opioid misuse. While the authors acknowledge this limitation (Bradford-Hill Criteria), it would be helpful to emphasize it further in the discussion to avoid misinterpretations.
- Methods/ Variable definitions: The "opioid misuse" variable is based on a single question from the YRBS, which reduces the depth of the analysis. A more comprehensive assessment tool or drug screening that included frequency, quantity, and routes of use would have been valuable. Some literature review regarding assessment tools and Drug screening
-
- Levy S, Minegishi M, Brogna M, Ross J, Subramaniam G, Weitzman ER. Screening for Nonmedical Use and Misuse of Prescription Medication by Adolescents. Subst Use Addctn J. 2025 Apr;46(2):357-363. doi: 10.1177/29767342241292419. Epub 2024 Dec 4. PMID: 39629781.
- Levy S, Brogna M, Minegishi M, Subramaniam G, McCormack J, Kline M, Menzin E, Allende-Richter S, Fuller A, Lewis M, Collins J, Hubbard Z, Mitchell SG, Weiss R, Weitzman E. Assessment of Screening Tools to Identify Substance Use Disorders Among Adolescents. JAMA Netw Open. 2023 May 1;6(5):e2314422. doi: 10.1001/jamanetworkopen.2023.14422. PMID: 37213103; PMCID: PMC10203888.
- Methods/ Variable definitions: Drug use among adolescents: The use of dichotomous variables for alcohol, marijuana, and depressive symptoms may oversimplify reality and lose nuances in risk patterns.
- Has the questionnaire been validated within the study population? What are the sensitivity and specificity? Has any validation been conducted using urine/blood tests to measure the use of opioids, nicotine, marijuana, or alcohol?
- Response rate and self-report bias: It would be advisable to mention the potential self-report bias of the YRBS (e.g., underestimation of opioid use due to stigma). The 57.5% response rate and its potential impact on representativeness could also be discussed.
-
- Jones CM, Clayton HB, Deputy NP, et al. Prescription Opioid Misuse and Use of Alcohol and Other Substances Among High School Students — Youth Risk Behavior Survey, United States, 2019. MMWR Suppl 2020;69(Suppl-1):38–46. DOI: http://dx.doi.org/10.15585/mmwr.su6901a5 https://www.cdc.gov/mmwr/volumes/69/su/su6901a5.htm?s_cid=su6901a5_w
- Multivariate analysis: Ordinal or multinomial logistic regression models could be explored to better capture the intensity of both PVE and opioid use, rather than dichotomizing the data.
- Social determinants and vulnerable groups: The article describes differences by sex, age, and race/ethnicity, but could provide more detail in stratified analyses by socioeconomic level, region/state level to help understand why certain groups show greater vulnerability.
- Spencer MR, Weathers S. Trends and risk factors of adolescent opioid abuse/misuse: understanding the opioid epidemic among adolescents. Int J Adolesc Med Health. 2020 May 12;33(4). doi: 10.1515/ijamh-2018-0179. PMID: 32396138.
- Data to action. Based on the results, the authors are poised to offer insightful recommendations that could not only enhance the methodological rigor of the survey but also significantly bolster national drug prevention programs aimed at adolescents.
Reviewer 2 Report
Comments and Suggestions for Authors
The topic of the paper is socially relevant primarily due to the increasing use of tobacco products among adolescents, particularly e-cigarettes. However, after reading the paper, it is considered important to review and analyze three important points:
- It is important to review the general objective and specific objectives (and clearly and consistently state in the different sections of the paper) the analysis performed using covariates and the study of the dose-dependent relationship between the frequency of current EVP use and prescription opioid misuse.
- The analysis o results section, the results, and the discussion should be consistent with the general and specific objectives (if, after analysis, it is considered to include them).
- Include in the conclusions everything related to the study of the dose-dependent relationship between the frequency of current EVP use and prescription opioid misuse.
Reviewer 3 Report
Comments and Suggestions for Authors
-
Authors are required to reduce the similarity rate, which is currently 24%.
-
In the keywords, an abbreviation was used; please write it in full.
-
The introduction is well written and contains all the elements of a pertinent introduction.
-
The Materials and Methods section is also good; however, please add the sample size.
-
Regarding depressive symptoms, to be more relevant and rigorous, it would be preferable to use a validated scalerather than relying only on 2 or 3 questions, which may not be sufficient.
-
In the Results, the sample size is very important and should be clearly reported as it reflects the representativeness of the study.
-
The Results section should be organized into subsections based on the type of data (e.g., age, sociodemographic characteristics). Please avoid headings such as “3.1. Figures, Tables and Schemes,” as there are no schemes included.
-
For better readability, add narrative comments before each table to introduce and explain the data presented.
-
Even though the sample size is important, the data presented are too brief and need to be expanded for better clarity and scientific value.
Reviewer 4 Report
Comments and Suggestions for Authors
This manuscript examined the association between electronic vapor product use and prescription opioid misuse among US high school students using 2021 YRBS data. They conducted adjusted Poisson regression analysis. This study showed that participants who used electronic vapor products had increased prevalence of opioid misuse.
Overall, this manuscript an important contribution to the literature for tobacco prevention and adolescent substance use prevention. The following are my comments that need to be addressed before considering for publication:
Abstract
- Please include age of the participants in the abstract.
- Please replace the word “non-user” and use person-first language throughout. For example, in this case you could say: “people who do not use EVPs”
- You use the terms such as “adolescents” and “teenager” interchangeably here. Please define exactly what you mean/age ranges of these different terms if you choose to use them interchangeably, or be consistent with your terms throughout the abstract and the manuscript (in the manuscript you also use “student” and “youth”.
Introduction
- Page 2 Line 51 – spell out United States here (as you do that throughout the rest of the manuscript
- Please go through the entire manuscript and fix to all person-first language. For example, please replace the word “smokers” and use person-first language throughout. For example, in this case you could say: “people who smoke.” Other words I noticed were “EVP users” and “opioid misusers.”
Methods
- Please explain why you used Poisson regression. Maybe I am misreading the methods, but it seems like your outcome of current EVP use is binary (yes/no) as well as prescription opioid misuse (yes/no). If that is the case, wouldn’t logistic regression be the most appropriate, since Poisson is meant for count variables? I would delineate more clearly exactly how you operationalized the e-cigarette use and opioid use in your regression model, as right now it is unclear to me.
Results
Results are clear and well organized.
Discussion
Great work connecting your results with other previously published research studies and writing an impactful conclusion.
Round 2
Reviewer 4 Report
Comments and Suggestions for Authors
The authors did a nice job responding to my suggestions. I just have one comment:
- On Page 5, you still use the term “EVP users” – please change to person first (e.g., people who use EVPs)
Author Response
Comment 1: On Page 5, you still use the term “EVP users” – please change to person first (e.g., people who use EVPs)
Response 1: Good catch, sorry to have missed that with it being part of the revisions. Fixed!